# Development Issues of Healthcare Robots: Compassionate Communication for Older Adults with Dementia

**DOI:** 10.3390/ijerph18094538

**Published:** 2021-04-24

**Authors:** Tetsuya Tanioka, Tomoya Yokotani, Ryuichi Tanioka, Feni Betriana, Kazuyuki Matsumoto, Rozzano Locsin, Yueren Zhao, Kyoko Osaka, Misao Miyagawa, Savina Schoenhofer

**Affiliations:** 1Institute of Biomedical Sciences, Tokushima University, Tokushima 770-8509, Japan; locsin50@gmail.com; 2Anne Boykin Institute, Florida Atlantic University, Boca Raton, FL 33431, USA; savibus@gmail.com; 3Graduate School of Health Sciences, Tokushima University, Tokushima 770-8509, Japan; tomotomochan0615@gmail.com (T.Y.); taniokaryuichi@gmail.com (R.T.); fenibetriana@gmail.com (F.B.); 4Graduate School of Engineering, Tokushima University, Tokushima 770-8506, Japan; matumoto@is.tokushima-u.ac.jp; 5Christine E. Lynn College of Nursing, Florida Atlantic University, Boca Raton, FL 33431, USA; 6Department of Psychiatry, Fujita Health University, Aichi 470-1192, Japan; taipeeyen@gmail.com; 7Department of Nursing, Nursing Course of Kochi Medical School, Kochi University, Kochi 783-8505, Japan; osaka@kochi-u.ac.jp; 8Department of Nursing, Faculty of Health and Welfare, Tokushima Bunri University, Tokushima 770-8514, Japan; miyagawa@tks.bunri-u.ac.jp

**Keywords:** artificial intelligence, geriatric care, healthcare robots, intelligent humanoid robots, natural language processing, older persons, robot compassion

## Abstract

Although progress is being made in affective computing, issues remain in enabling the effective expression of compassionate communication by healthcare robots. Identifying, describing and reconciling these concerns are important in order to provide quality contemporary healthcare for older adults with dementia. The purpose of this case study was to explore the development issues of healthcare robots in expressing compassionate communication for older adults with dementia. An exploratory descriptive case study was conducted with the Pepper robot and older adults with dementia using high-tech digital cameras to document significant communication proceedings that occurred during the activities. Data were collected in December 2020. The application program for an intentional conversation using Pepper was jointly developed by Tanioka’s team and the Xing Company, allowing Pepper’s words and head movements to be remotely controlled. The analysis of the results revealed four development issues, namely, (1) accurate sensing behavior for “listening” to voices appropriately and accurately interacting with subjects; (2) inefficiency in “listening” and “gaze” activities; (3) fidelity of behavioral responses; and (4) deficiency in natural language processing AI development, i.e., the ability to respond actively to situations that were not pre-programmed by the developer. Conversational engagements between the Pepper robot and patients with dementia illustrated a practical usage of technologies with artificial intelligence and natural language processing. The development issues found in this study require reconciliation in order to enhance the potential for healthcare robot engagement in compassionate communication in the care of older adults with dementia.

## 1. Introduction

Communicating with others is a critical aspect of people’s lives, required for actively engaging in society, passing on wishes and needs, and sharing attitudes, knowledge and experiences with other members of the community [1]. Human communication is essential in enhancing social and perceptive interactions among older people [2], and social engagement can help people cope better with the effects of aging, delaying the development of symptoms of dementia [3].

Often, people with dementia experience symptoms such as agitation, which can result in increasing the care burden, and intellectual stimulation may benefit these people by improving their understanding, emotional and social well-being and the performance of Activities of Daily Living [4]. Studies have suggested that the continuous enrichment of social interaction is also necessary for the maintenance of cognitive functioning among older people.

Japan had roughly 1.9 million nursing care workers as of 2016, and Health, Labor and Welfare Ministry estimates showed that with a growing demand for staff by the rapidly aging population, the country faces a shortage. The current pace of the increase in the care-staffing practice shows a shortage of 337,000 nursing care workers by 2025 [5].

Other developed countries are confronting the reality of a nursing workforce that is rapidly aging, giving priority to the development of healthcare robots (HCRs) to assist in the care of older adults in the developed world, e.g., Japan [6], the United States [7], Italy [8], and Scandinavia [9]. Although HCRs are becoming widely accepted as integral to many aspects of the physical and psychosocial care of the growing population of older adults, much development work is needed to maximize the impact of HCRs on this vulnerable population [10]. One area that demands critical developmental attention is designing compassionate HCRs to possibly solve workforce shortages in older adult healthcare settings.

Compassionate care in clinical healthcare refers to the actualization of acknowledgment, engagement and action in response to patient suffering [11]. Dewar and Nolan [12] considered compassion in the care of older people as having four elements: (1) a relationship based on empathy, emotional support and efforts to understand and relieve the patient’s distress and suffering; (2) effective interactions between staff and patients; (3) being active participants in decision making; and (4) contextualized knowledge.

Developing compassionate care for older people begins with the core principle of knowing clearly what the relationship with the patient is about. This process of constructing meaning in the interaction and conversation can be achieved by asking questions of the older adults, having conversations with them, enquiring into what and who are important to them and what matters to them [13]. Especially for older people with dementia, it is important to communicate in skillful and sensitive ways, providing emotional support and necessary stimulation, while expressing reassurance and security through appropriate verbal and non-verbal approaches [14].

Scholars and practicing geriatric nurses, geriatric psychiatrists and the public must always recognize that compassion remains an integral element in human caring, even despite several technological advancements in geriatric care practice [15]. While all healthcare providers should demonstrate compassion and ethical behavior, there are times when lapses in compassion and conscience can result in compromised patient care, and even serious life-threatening incidents. Thus, HCRs must be designed to enhance humanness, respect the autonomy of older adults and demonstrate compassion through technological programming with compassionate language that enables the HCR to express the caring behavior [10].

HCRs are robots used in the healthcare setting [16], although their functions and performance capabilities are still under development [17]. Currently, HCRs do not possess the competencies of expressing “caring in nursing” that a human nurse expectedly shows [16].

For these purpose, an HCR should be able to provide interpersonal communication that can understand, empathize and share empathy with patients. In order for an artificial intelligence (AI)-equipped HCR to have a “heart/mind” that is capable of compassionate communication, it requires the ability to first observe the patient’s need, to correctly evaluate it and to communicate its findings to the patient in appropriate words. For example, a series of actions that integrate not only words but also “knowledge, judgment, technical skills, and care” are expected of these robots [18]. The AI-equipped HCR will be programmed to appreciate the meaning of the caring experience. When these robots can “express” themselves with human-like emotive behaviors, they will be able to convey empathic understanding to the patients and their families [19].

Effective communication is imperative to ensure quality of care for people living with dementia [20]. However, language performance is influenced both by normal aging and by the development of dementia. Thus, it is critical that the person with dementia is not stressed in attempting to respond during conversation, and it is important to avoid confrontation and conflicts with people suffering from dementia [21]. Furthermore, many aspects of the daily lives of older adults are associated with hearing ability, showing that hearing loss affects the quality of life, social relationships, motor skills and psychological aspects, as well as the function and morphology of specific brain areas [22]. For success in compassionate communication activities, it is important that older adults can enjoy interactive engagements with the HCR [17].

HCRs used in Japanese hospitals and healthcare facilities often function with low-fidelity capabilities. Intermediaries are proposed in these situations [23]. It was found that human intermediaries are indispensable in situations with inadequately programmed communicative robots, particularly those interacting with older persons [24]. The involvement of intermediaries is necessary to obtain adequate evidence that can be used to fine-tune programs to enable robots to provide compassionate care [19,25].

In the evolution of HCRs, one challenge is to develop AI that can anticipate the next move of an individual person [26]. Although it may be possible for humans to instruct humanoid robots to perform medical or nursing care behaviors, no database for decision-making exists as a basis for instructions [27]. Therefore, developing an AI-based database requires further study, particularly emphasizing compassionate communication between persons and HCRs [28]. In a previous study [24,29] involving older persons, observations of conversations were evaluated to determine whether or not conversations between intelligent machines and older persons were meeting the goals of the activity. In the robot’s NLP program (*Kenko Okoku* Talk application) [29], which was used to make the robot converse with humans, the robot’s database did not include a ‘wait time’ for the interval between responses, making it difficult for the robot to continue the conversation.

The purpose of this case study was to explore the developmental issues affecting healthcare robots in the area of compassionate communication with older adults with dementia.

## 2. Materials and Methods

### 2.1. Description of Pepper—A Healthcare Robot with a Developed/Installed Application Program

Pepper is a humanoid robot developed by Softbank Robotics that has been available since 2014 [30]. SoftBank Robotics, formerly Aldebaran Robotics, has been providing in-service education for medicine and nursing care where needed [31,32,33]. In this study, a new conversation application was used for the communication process between Pepper and the subjects who participated, applying the Wizard-of-Oz (WoZ) technique, in which an operator remotely operated and controlled the robot, including its speech, gestures, and navigation [34].

The application program for an intentional conversation was jointly developed by Tetsuya Tanioka’s team and the Xing Company. This program allows Pepper’s words and head movements to be remotely controlled by connecting it to two computer tablets (iPad minis) using a Wi-Fi connection. To ensure the timely input of conversational texts, this application was uploaded to the tablet using a keyboard through a Bluetooth connection. This allowed the operator to directly input the words for the conversation that he or she has in mind. This system also had a switch that made Pepper’s head and face move towards the person who was speaking. Additionally, this program included template responses, questions and replies that fit the conversation (Figure 1).

### 2.2. Comparison of Application Programs for Intentional Mutual Conversations

Based on an exploratory case study conducted by Dr. Tanioka’s team using Pepper’s conversation application (*Kenko Okoku* Talk) [29] and a new application program for intentional mutual conversation developed by Dr. Tanioka’s team (Figure 2), several distinctions were identified, as summarized in Table 1.

### 2.3. Procedure for Data Collection

An exploratory descriptive case study was conducted to provide explanations of conversational engagements with Pepper robots. The observed data consisted of participant observer reports regarding their objective annotations of the human–robot engagements, and field notes. Based on the audio-visual recordings, the conversations between Pepper and the subjects were analyzed, initially transcribing the conversation, and using a highlighting technique, the words, phrases and sentences were identified. These observed data and qualitative descriptions and considerations provided the results of the study.

Researchers ensured that the subjects adhered to the study procedure as they interacted with Pepper. They also provided key advice, giving definite, concrete and explicit instructions. The healthcare provider acted as an intermediary, assisting in the conversation between Pepper and the subjects when needed. The length of the conversation time with the Pepper robot was about 20 to 30 min.

Two subjects were included in this case study. This was the first time the subjects had participated in a conversation with the Pepper humanoid robot. The inclusion criteria were as follows. Subjects with dementia were selected based on their ability to converse in the Japanese language. Dementia was diagnosed using Hasegawa’s Dementia Scale–Revised (HDS-R). With an HDS-R score of 20 or less, dementia was suspected (30 is the highest possible score and can be interpreted as “no dementia suspected”). More specifically, scores on the HDS-R include the severity of dementia: scores between 20 and 15 points are categorized as mild, those between 14 and 10 points are moderate, scores between 9 and 5 points are moderately severe, and scores below 5 points are designated as severe [35]. Subjects did not have clinical psychiatric changes in their symptoms such as those reflecting dementia or acute decompensation of concomitant systemic diseases and were stable in their pharmacological treatment in the last 3 months.

Exclusion criteria: subjects under treatment for alcohol or drug abuse, aggressive or with violent behavior, having a terminal illness, severe co-morbidity, severe auditory impairment, inability to understand Japanese or a lack of informed consent were exclusion criteria for this study. Data were collected in December 2020.

Subjects A and B were women in their 90s, diagnosed with dementia.

Subject A was diagnosed with Alzheimer’s disease (G30). Her HDS-R score was 7 points; she was diagnosed with moderately severe dementia. Her medications were amlodipine besilate 2.5 mg/day, tofogliflozin hydrate 20 mg/day and glimepiride 1mg/day. She was able to walk on crutches.

Subject B was diagnosed with vascular dementia (F01.5). Her HDS-R score was 12 points, with a diagnosis of moderate dementia. Her medications were amlodipine besilate 2.5 mg/day and apixaban 5 mg/day. She was able to walk with a walker.

### 2.4. Setting of the Case Study

The location of the study was a rehabilitation area at a nursing care facility for people with dementia. Two rooms in this area were prepared for the study—the room where the conversations with the Pepper robot took place and the robot operator room. The exploratory case study was set up based on the situation of a conversation between Pepper robot and two female subjects. In the separate operators’ room, one operator and one assistant were assigned to input phrases remotely into the Pepper robot’s conversation application program via a tablet and keyboard. The operator’s assistant aided the operator in inputting the Pepper robot’s actions, such as acknowledgment gestures and nodding movements during the conversation. The operator was a co-researcher who was a nurse and had clinical research experience involving the Pepper robot.

One camera was placed in the operator’s room to record the interaction between the operator and assistant, and to record the situation while operating Pepper (Figure 3). The interactions between Pepper and the two subjects were recorded using three digital cameras and two intermediary observers with field notes, noting significant events occurring during the conversation.

### 2.5. Ethical Considerations

Approval from the Ethics Committee of Tokushima University Hospital (#3046) and from the Mifune Hospital Clinical Research Ethics Review Committee (#201180502) was obtained. All participants gave their informed consent before inclusion in the study. The purpose and methods used in the study were explained to all subjects and their guardians. Subjects and their guardians were assured that their personal information was protected, that the results would be reported as an aggregate, and that the data generated would be used only for research purposes.

## 3. Results

### 3.1. Accurate Sensing Behavior for “Listening” to Voices Appropriately and Accurately for Interaction with Subjects

Pepper’s conversation began after greeting the participants with “Hello.” The contents of the conversation were about eating, how to spend the New Year, and their previous work. Both Subjects A and B had presbycusis, and it was difficult for them to hear Pepper’s utterances (Figure 4).

Thus, a care worker acting as an intermediary conveyed what Pepper was saying by whispering in the subjects’ ears. As a result, both subjects A and B could understand and proceed with the conversation. Both Subjects A and B had a low voice volume, and it was difficult for the operator to hear the subjects’ voices, so a microphone was used in this human–robot interaction session.

### 3.2. Inefficiency in “Listening” and “Gaze” Activities

When Pepper told Subjects A and B to “Sing Japanese nursery rhymes,” they sang nursery rhymes.

Pepper continued to dialogue with subject A; however, when it was time to end the conversation with subject A, Pepper was not looking at Subject A, but was instead looking toward Subject B. During this interaction time, Pepper’s eye contact was not on Subject B at all, but she continued to talk.

### 3.3. Fidelity of Behavioral Response

Pepper asked Subject B “What did you eat for lunch?”

She replied, “I forgot! I am sorry.”

Pepper asked, “What do you plan to do on New Year’s Day?”

She replied, “On New Year’s Day, I will play Hanetsuki (battledore).”

Pepper asked her, “May I have a New Year’s gift?” and she replied, “Yes, I would be happy to give you one.”

However, the subject’s speech was unclear. Therefore, the intermediary had to restate what she said.

### 3.4. Deficiency in Natural Language Processing AI Development, i.e., the Ability to Respond Actively to Situations That Were not Pre-Programmed by the Developer

When the conversation ended and the older persons were able to interact at a closer distance, listening was clearer, and the conversation seemed to transpire much more smoothly.

After the conversation, both Subjects A and B stroked Pepper’s head and repeatedly thanked Pepper, saying, “Thank you.” At the end of the conversation with Pepper, the older people exhibited a sense of familiarity with Pepper. They touched the Pepper robot and reluctantly parted with Pepper (Figure 5). From this observed communicative-interactive process, the evidence showed that Pepper succeeded in entertaining these older people through their interactions.

## 4. Discussion

### 4.1. Accurate Sensing Behavior for “Listening” to Voices Appropriately and Accurately for Interaction with Subjects

Due to the low volume of the older subjects’ voices and the distance observed during the conversation, it was difficult for Pepper to show accurate sensing behavior to “listen” to the voices of the subjects.

Voice plays a large part in the communication interaction between Pepper and the subjects. Since the subjects’ voice volume was low, a headset (earphone and microphone) was considered necessary. In the case of older patients with symptoms such as dementia and deafness, relatively smooth dialogue can be achieved with the intervention of a care worker, such as a nurse or an occupational therapist who plays a supporting role [23]. This role supported the practice of having a human intermediary as a result of the current program limitations of the Pepper robot. Nevertheless, if a human support (intermediary) role [23] is always required for an autonomous interactive robot, the cost of rehabilitation for older people using the robot will increase. To deal with such patients, it is necessary for robots to have a function assuring the highly accurate recognition of the voice of older adults and a function of making the voice of the robot easy to hear, even for hearing-impaired patients.

### 4.2. Inefficiency in “Listening” and “Gaze” Activities

The human–robot interaction illustrated a listening and gaze problem. As there were two subjects in the conversation with Pepper, it could only visually engage with one of the intended subjects at one time. This seemed to demonstrate that the robot’s programming and its operation needed to have better coordinated response behaviors. The other subject was seemingly “ignored.”

In the case of healthy people, the timing (pace) of conversation and the line of sight during the conversation are important factors. A previous study demonstrated that human–robot gaze interactions conform to phenomena observed in human–human interaction [36]. Eye contact with a robot can elicit similar types of automatic affective and attentional reactions as eye contact with another human being. This ability might enhance pleasant experiences in social interaction between humans and robots, which in turn might facilitate robots’ integration into human societies [37]. For robots to express compassionate communication with older adults, future development needs to consider improving the robot’s ability to communicate with more than one person or to a group of people, while maintaining appropriate eye contact/gaze with the person talking.

### 4.3. Fidelity of Behavioral Response

With the operator inputting behavioral operational programs from the computer keyboard, Pepper robot’s responses may appear to be activities devised by the operator who was communicating with the subjects. However, essentially it is the Pepper robot’s programming and operational capabilities using NLP that allow it to voice verbal responses. The AI for NLP development, as a hypothetical example of human-to-nonhuman mutual conversation, can be further examined. This is because a communication program has yet to be developed with AI and databases for NLP that will provide expressive emotional behavior, as is the case for human compassion.

### 4.4. Deficiency in Natural Language Processing AI Development, Ability to Respond Actively to Situations That Were Not Pre-Programmed by the Developer

The ability to express compassion (affective artificial compassion [38]) is acknowledged as the least developed area of AI. A developmental issue of HCRs in relation to compassionate communication is NLP AI development, i.e., the ability to respond actively to situations that were not pre-programmed by the developer.

In human–robot interactions, empathy and compassion are acknowledged as key factors in overcoming the limitations of contemporary HCRs [39]. Most clinical work and patient care applications require much more cognitive adaptability and problem-solving communication skills than is possible for a regular computer [40]. HCRs are envisioned to deliver meaningful benefits through effective interaction with humans to fulfil their expectations [41]. In contrast to automation, which follows pre-programmed “rules” and is limited to specific actions, autonomous robots are required to have a context-guided behavior adaptation capability, which would allow them to have a degree of self-governance [42]. It should enable them to learn and respond actively to situations that were not pre-programmed by the developer. To establish a successful human–robot interaction using compassionate natural language, aside from perceptual and cognitive capabilities, an autonomous robot should be able to adapt its behavior in real time and often in partially unknown settings, making necessary adjustments to the situation at hand [43].

## 5. Conclusions

In this article, we explored the development issues of HCRs in regard to compassionate communication with older adults with dementia. Our findings revealed four development issues: (1) accurate sensing behavior in order to “listen” to voices appropriately and accurately for the purpose of interacting with subjects; (2) inefficiency in “listening” and “gaze” activities; (3) fidelity of the behavioral response; and (4) deficiency in natural language processing (NLP), AI development, i.e., the ability to respond actively to situations that were not pre-programmed by the developer. Faced with increasing shortages of healthcare workers for the burgeoning population of older adults, the healthcare sector should acknowledge that beneficial communication opportunities for this population are increasingly scarce. If interactive robots are used to care for older people, it is urgent to create a database and AI for NLP to achieve this. Considering that one of the expected functions of HCRs is as a communication partner for older people, further improvements are needed in order for them to have the abilities necessary to engage in behaviors depicting compassionate conversational activities.

## Figures and Tables

**Figure 1 ijerph-18-04538-f001:**
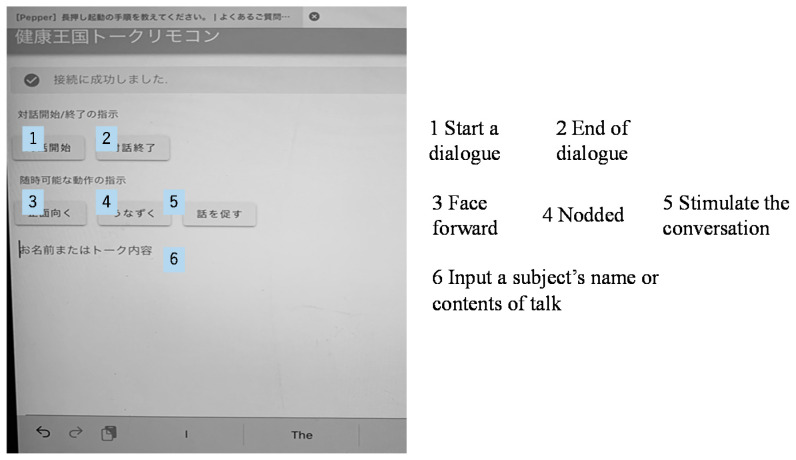
The display of the iPad mini remote control of the application program for an intentional conversation using the Pepper robot.

**Figure 2 ijerph-18-04538-f002:**
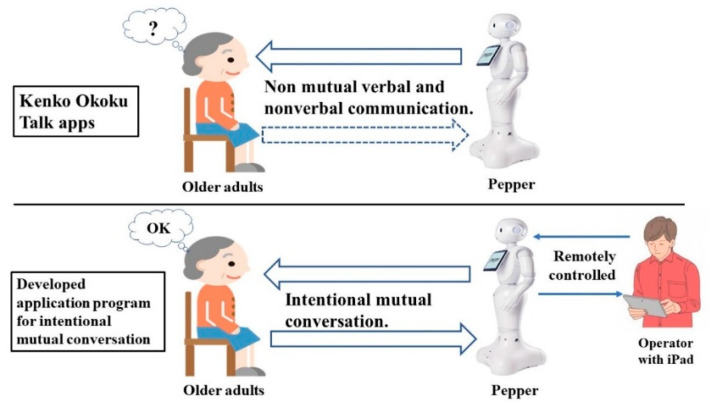
Difference between the Pepper conversation application (*Kenko Okoku* Talk) and the application program developed for intentional mutual conversation.

**Figure 3 ijerph-18-04538-f003:**
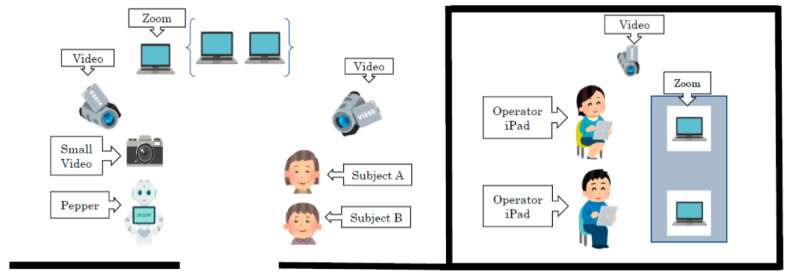
The situation of Pepper interacting with subjects.

**Figure 4 ijerph-18-04538-f004:**
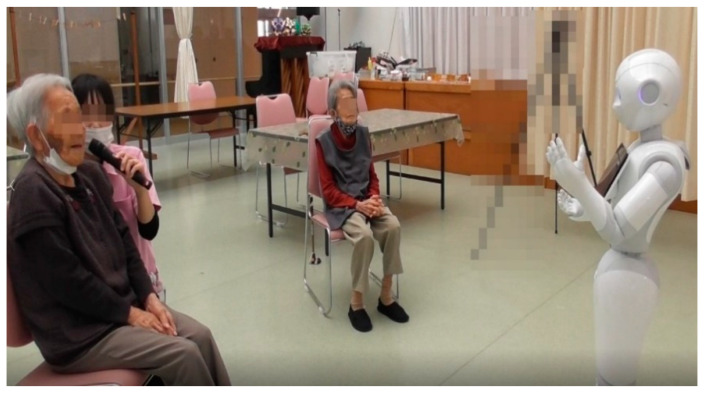
Subjects A and B, who were women in their 90s with Alzheimer’s disease.

**Figure 5 ijerph-18-04538-f005:**
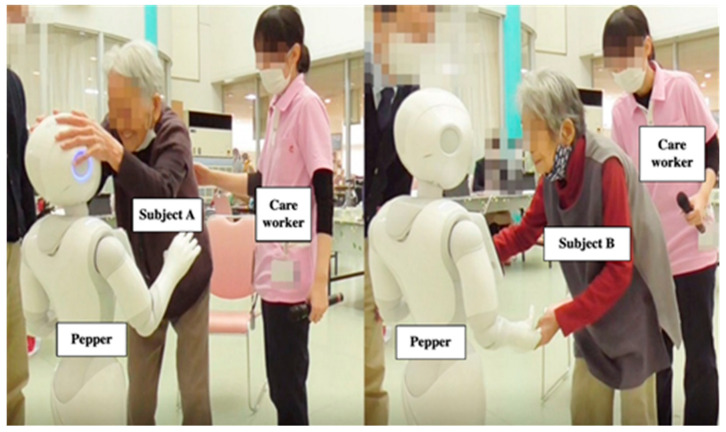
Subjects A and B stroked Pepper’s head and repeatedly thanked Pepper, saying, “Thank you.”

**Table 1 ijerph-18-04538-t001:** Comparison of *Kenko Okoku* Talk and the application program developed for intentional mutual conversation.

Application	*Kenko Okoku* Talk	Application Program for Intentional Mutual Conversation
Robot body	Pepper	Pepper
Features	Robot is able to	This new system allows:
	ask questions to subject	The voicing of robot to be organized based on the situation
	answer question	Response time can be managed according to the conversation situation
	explain about certain topics	Conversation content can be inputted and managed in real time Engaging the robot and subject within a more ‘natural’ conversation Robot‘s head and face can be moved towards the conversation partner System includes template responses, questions, and replies that fit the conversation Enable the robot to communicate with multiple conversation partners
Problems	1. Asynchronous timing response, e.g., robot moves to the next topic while subject is still talking 2. The robot data base does not include ‘wait time’ for the interval between responses accurately 3. Robot‘s speed and the emotional “color” are difficult for the older persons to follow 4. Eye contact cannot be managed 5. Unable to communicate with multiple conversation partners	Needs human operator(s) to input the content and control the conversation in a timely manner and with appropriate “interval time”.

## Data Availability

The data presented in this study are available on request from the corresponding author. The data are not publicly available due to privacy and ethical restrictions.

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
