# Peer review of "Development Issues of Healthcare Robots: Compassionate Communication for Older Adults with Dementia"

_ijerph, 2021, doi:10.3390/ijerph18094538_

Round 1
Reviewer 1 Report
no comments
Author Response
Thank you very much.

Reviewer 2 Report
Introduction
Line 36 – “ illustrated the use of technologies with Artificial Intelligence and Natural Language Processing…” what did this study illustrate, did it illustrate that is a potential for future use, or that there is still problems. The sentence needs attention.
Line 49-50 – How do you suddenly go from improved healthcare to robots, how does the robots fit into the bigger picture? Why is it important to design health care Robots?
Line 60 – “critical aspect of a persons” The letter “a” is needed.
Line – “states in dementia, borrow heavily”
Line 77 – “engaged, exercise their cognitive”.
Line 78 – “language, and facilitating the development”
Line 102 – “the next move of human persons” – Maybe instead , the next move of a human or an individual..
- The introduction is overall well written. I however believe that the paper needs to be send to a language editor as there are various grammar problems with the paper.
- Give more attention at the start of you suddenly move from nurses to robots, restructure the intro to provide a better flow to your argument.
Methods
Line 136-137 – “Dewar and Nolan [28] 136 considered compassion in older people care setting as having four elements” – Sentence needs attentions, please rewrite.
Line 127-182 – This section reads like a second introduction to AI robots and the Pros and cons of theses Robots. There is no material or methods being addressed here. Instead, you keep motivating why Robots are needed?? Rather combine this with your introduction.
Line 191 – “navigation, etc.” – Please do not use etc., as this is your method, and the reader must fully understand what was done and how it was performed.
Line 241 – Only 2 subjects used? I do not believe that any conclusion can be derived from only two subjects. Secondly, did these subjects also made use of the previous model of Pepper to conclude that the newer model and system has a better result? How old were the subjects?
Line 226 – “two female subjects.” This was only performed on females; would it not be better to test the system on males and females in order to determine if both sexes react the same to the robot interaction?
- How was the data collected in term of Qualitative or quantitative research? You indicated that it was video recorded. What happed after that, how was the situation “scored” or analysed? Was notes taken and how and how was it interpreted?
Line 286 - Case Example – Move this part earlier, in order to first understand who, the subjects were, how many and then only to know what was done to them.
Were the operators of the Robot trained? Do they need a specific qualification to understand the medical field of work in order to understand how and when to react?
Results
Line 306-308 – “A care worker who was an intermediary conveyed what Pepper was saying by whispering in the subjects' ears. As a result, both subjects A and B were able to understand and proceed with the conversation.” – This could have affected the results, as you wanted to test how the interaction were with the robots and the participants, without any other human influence. In this case they still had a human influence that they recognised and were comfortable with.
- The results are vague and needs to be structured better.
Discussion
- As your results were not well structured, it carried over to the discussion section. Only one or two vague results of your study were Discussed. Instead, you provided more background of the literature and what is important regarding communication and compassion.
- I believe the discussion should have focused on what was seen, why it might have happened (reasons) and what the possible solutions would be. As well as were there improvements with regards to the previous Robot…
Overall
- This paper needs more structure especially to the methods, results and discussion.
A Question that is raised is, that AI is intelligence without the need for a human to physically be involved in the process. In you robot a human is actively controlling the robot and provide the input necessary? Can that still be seen as AI? I do not believe so.
Author Response
Response to Reviewer 2 Comments
Point 1: Introduction: Line 36 – “illustrated the use of technologies with Artificial Intelligence and Natural Language Processing…” what did this study illustrate, did it illustrate that is a potential for future use, or that there is still problems. The sentence needs attention.
Response 1: The sentence was revised based on your suggestion (line 38-39).
Point 2: Line 49-50 – How do you suddenly go from improved healthcare to robots, how does the robots fit into the bigger picture? Why is it important to design health care Robots?
Response 2: The sentence was revised based on your suggestion (line 59-70).
Point 3: Line 127-182 – This section reads like a second introduction to AI robots and the Pros and cons of theses Robots. There is no material or methods being addressed here. Instead, you keep motivating why Robots are needed?? Rather combine this with your introduction.
Response 3: The sentence was revised based on your suggestion. That part was combined with introduction (line 71-110).
Point 4: Line 60 – “critical aspect of a persons” The letter “a” is needed.
Response 4: The sentence was revised based on your suggestion.
Point 5: Line 70 – “states in dementia, borrow heavily”
Response 5: The sentence was revised: this sentence was deleted.
Point 6: Line 77 – “engaged, exercise their cognitive”.
Response 6: The sentence was deleted.
Point 7: Line 78 – “language, and facilitating the development”
Response 7: The sentence was deleted.
Point 8: Line 102 – “the next move of human persons” – Maybe instead, the next move of a human or an individual.
Response 8: This sentence was deleted.
Point 9: The introduction is overall well written. I however believe that the paper needs to be send to a language editor as there are various grammar problems with the paper.
Response 9: Thank you very much for the suggestion. This manuscript has been edited by the language editor.
Point 10: Give more attention at the start of you suddenly move from nurses to robots, restructure the intro to provide a better flow to your argument.
Response 10: The sentence in the introduction was revised based on your suggestion.
Point 11: Methods: Line 136-137 – “Dewar and Nolan [28] 136 considered compassion in older people care setting as having four elements” – Sentence needs attentions, please rewrite.
Response 11: The sentence was revised and re-written based on your suggestion.
Point 12: Line 191 – “navigation, etc.” – Please do not use etc., as this is your method, and the reader must fully understand what was done and how it was performed.
Response 12: The abbreviation “etc.” was removed.
Point 13: Line 241 – Only 2 subjects used?
Response 13: Thank you for your question. Yes, the study is a case using two subjects as participants.
Point 14: I do not believe that any conclusion can be derived from only two subjects.
Response 14: The conclusion was derived from the findings of the study. However, rather than the conclusion being a basis for a prescription or prediction, as a case study the conclusion focused on description and explanation of the findings. The purpose of the case study was to determine the development issues. Thank you very much.
Point 15: Secondly, did these subjects also made use of the previous model of Pepper to conclude that the newer model and system has a better result?
Response 15: Thank you very much. The study did not intend to compare the results of using the ‘previous model of Pepper” and its “newer model and system”. This section was revised.
Point 16: How old were the subjects?
Response 16: The subjects were in their 90s. This information is stated in line 225.
Point 17: Line 226 – “two female subjects.” This was only performed on females; would it not be better to test the system on males and females in order to determine if both sexes react the same to the robot interaction?
Response 17: Thank you for your suggestion. In this case study, we had two female subjects who met the inclusion criteria. In the future study, we will involve male subjects and determine if gender influences human-robot interactions.
Point 18: How was the data collected in term of Qualitative or quantitative research? You indicated that it was video recorded. What happed after that, how was the situation “scored” or analysed? Was notes taken and how and how was it interpreted?
Response 18: Thank you very much for your questions.
We do not have quantitative data. Observed data consisted of participant observer reports regarding their objective annotations of the human-robot engagements, supported by their field notes. From the audio-visual recordings, the conversation between Pepper and the subjects were analyzed, initially transcribing the conversation, and using highlighting technique of the words, phrases, and sentences. From these data qualitative descriptions were provided.
Point 19: Line 286 - Case Example – Move this part earlier, in order to first understand who, the subjects were, how many and then only to know what was done to them.
Response 19: Thank you for your suggestion. That part was moved to the section on procedure, subjects (line 225-232), and data collection section (line 199-204).
Point 20: Were the operators of the Robot trained? Do they need a specific qualification to understand the medical field of work in order to understand how and when to react?
Response 20: The operator was a co-researcher who is a nurse and is an experienced clinical researcher, particularly on using Pepper robot. This information was added in lines 242-243.
Point 21: Results: Line 306-308 – “A care worker who was an intermediary conveyed what Pepper was saying by whispering in the subjects' ears. As a result, both subjects A and B were able to understand and proceed with the conversation.” – This could have affected the results, as you wanted to test how the interaction were with the robots and the participants, without any other human influence. In this case they still had a human influence that they recognised and were comfortable with. The results are vague and needs to be structured better.
Response 21: Thank you very much for this comment. However, the purpose of the study was not to test how human-robot interaction can occur independently. This was therefore clarified in the revision.
In the introduction, the authors pointed out that because the Pepper robot's performance particularly regarding language is low, the current robot function requires the role of an intermediary.
This situation identified the deficiency of Pepper robot programming such as the limitation of its “autonomous” conversational abilities. Because of this limitation, the input provided clarified the important role of an intermediary - a person who is able to simplify the specifics of language in order for the subjects to more clearly understand the verbalization of the humanoid robot during the robot-human conversation.
Point 22: Discussion: As your results were not well structured, it carried over to the discussion section. Only one or two vague results of your study were discussed. Instead, you provided more background of the literature and what is important regarding communication and compassion.
Response 22: Thank you very much for your comments. In the discussion, we added discursive data from results of the observations, and the qualitative descriptions based on the conversations between Pepper and the subjects. We added observational data from what were observed in the video recordings.
Point 23: I believe the discussion should have focused on what was seen, why it might have happened (reasons) and what the possible solutions would be. As well as were there improvements with regards to the previous Robot…
Response 23: Thank you very much.
The discussion was revised focusing on the reasons to explain and describe the findings and identified possible solutions.
However, your comment on including “improvements with regards to the previous Robot,” cannot be addressed as this was not a comparative study. Only one Pepper robot was used. Nevertheless, we have modified this section.
Point 24: Overall: This paper needs more structure especially to the methods, results and discussion.
Response 24: Further revisions were made to the data discussion and other aspects concerning the study in order to provide a clearer and more wholesome presentation of the findings of the case study.
Point 25: A Question that is raised is, that AI is intelligence without the need for a human to physically be involved in the process. In your robot, a human is actively controlling the robot and provide the input necessary?
Response 25: Thank you very much for this question. As provided in the Introduction and the description of the limited capabilities of the Pepper robot programing, language nuances could not be captured effectively, thereby requiring a human “intermediary” to clarify and simplify the interactive language content. This feature supported the practice of having human intermediary consequent to the current program limitations of Pepper robot.
Point 26: Can that still be seen as AI? I do not believe so.
Response 26: Yes, it is not AI. With the operator inputs from the computer keyboard, while it may appear that it is the operator who is communicating with the subjects, essentially it is Pepper robot’s text-to-speech (TTS) system converting normal language texts into speech. In the future, we think that communication programs with AI and databases for NLP will be developed further in order to provide HCRs the expressive emotional behavior, much like those conveyed by human beings in expressing compassion.
This paper presents the progress of the AI development supporting the new application program for intentional compassionate conversation that allows Pepper to be remotely controlled, in addition to its default mode. Therefore, an operator took part to provide the input for the conversation. As the reviewer pointed out, what the operator is uploading is not AI. With the operator inputting from the keyboard, it is the operator that is communicating with the subjects using Pepper robot’s AI capabilities using natural language processing. This suggests a future development of the influence of NLP programming and caring database for Pepper robot communicative proficiency.

Round 2
Reviewer 2 Report
Thank you for the corrections and amendments. Although there are only two participants that was used in this study, I do believe that these results are important to be published.
This manuscript is a resubmission of an earlier submission. The following is a list of the peer review reports and author responses from that submission.
Round 1
Reviewer 1 Report
Title: Development Issues of Healthcare Robot Artificial Intelligence for
Rehabilitation and Recreational Activities of Older Adults with Dementia and
Chronic Mental Illness Through Cases Focused on Artificial Intelligence and
Natural Language Processing
Thanks for the invitation in reviewing this interesting paper and this study address an important area for older adults with dementia and patient with chronic mental illness.
My comments to this manuscript as follow:
- Although the idea is important, the title is too complicated for readers to follow. Moreover, chronic mental illness shouldn’t be a proper term to use. Theoretically, we can prevent the cognitive / socio-cognitive function decline in people with dementia, but the disease “dementia” cannot be prevented too.
- The abstract is over-stretched. For example, in the first sentence “To prevent dementia, increased communication and interdisciplinary collaborative approach is required.” … but how ? Authors show revise with more solid background for readers to follow. I suppose this study share the importance of AI and communication activities with robot ?
- The abstract is not in a proper format. Please check the format as stated in authors’ guideline
- Authors jumped their ideas to “social isolation and social interaction” in introduction, but they stressed the importance of “communication” in the abstract? Their ideas were not well-organized. Suggest authors to restructure the introduction in making a more coherent writing for readers’ easier understanding.
- Authors should clearly explain why older adults with dementia and chronic mental illness were grouped together as their target participants. Clinically, these two groups are very different. This should be properly address.
- Some easily observable editing mistakes in line 84-85.
- Authors should make clear what is the Framework of Compassionate Humanoid Health care Robots Communication and what paradigm it belongs to ? what level of evidence had been yielded? And most importantly, any support of application of this framework to older adults with dementia and chronic mental illness.
- Suggest authors should check for similarity (by either Veriguide orTurn-it-in), as I noted the writing style varies over the entire manuscript.
- Please write clearly how many subjects were recruited for this study. In line 32, authors marked 2, but in line 229, authors stated 6.
- Please write clearly if Pepper will interact with one subject or one group of subjects in the stated procedure ??
- How can authors ensure the compliance and adherence of subjects when they interact with Pepper? This is a very questionable confounding factor.
- Base on what inclusion and exclusion criteria did authors select their subjects ? what types of schizophrenia ? how chronic she was ?? based on what criteria that you chose ?? very unclear.
- What level of dementia did the subjects E and F were ? unclear too.
- Where is the results ?? It seems this section is omitted in the entire manuscript.
- Authors should majorly revamp their discussion. The discussion in its current form just like another part of introduction. Suggest authors should follow the track of writing from Background (what knowledge had been identified and what knowledge gap should be addressed by authors), then Aim (I suppose they are working on Communication ?!). then Methods ( with Peppers and AI ??), then Results (Even this is a case study, authors should report results in participants), then Discussion (discussion with result collected)
- Based on the present manuscript, I cannot seem how authors integrate AI and NLP at all. Can author reinforce this part in their methods ?? Moreover, the conclusion is over-stretched with consideration of all the weakness as I mentioned from point 1-15.
- To enrich the readability of the manuscript, English editing is a must for the revision.
Author Response
Point 1: Although the idea is important, the title is too complicated for readers to follow. Moreover, chronic mental illness shouldn’t be a proper term to use. Theoretically, we can prevent the cognitive / socio-cognitive function decline in people with dementia, but the disease “dementia” cannot be prevented too.
Response 1: Thank you for your comment. We revised the title into:
Developmental Issues of Healthcare Robots and Compassionate Care in Rehabilitation and Communication Activities of Older Adults with Dementia and Persons with Schizophrenia.
Point 2: The abstract is over-stretched. For example, in the first sentence “To prevent dementia, increased communication and interdisciplinary collaborative approach is required.” … but how? Authors show revise with more solid background for readers to follow. I suppose this study share the importance of AI and communication activities with robot?
Response 2: Abstract and background were revised to describe the importance of AI and communication activities with robots. Please see the revised abstract in the manuscript file (Page 1). It shows red font.
Point 3: The abstract is not in a proper format. Please check the format as stated in authors’ guideline.
Response 3: Abstract was revised following the guideline. Please see the revised abstract in the manuscript file (Page 1).
Point 4: Authors jumped their ideas to “social isolation and social interaction” in introduction, but they stressed the importance of “communication” in the abstract? Their ideas were not well-organized. Suggest authors to restructure the introduction in making a more coherent writing for readers’ easier understanding.
Response 4: We revised and restructured the introduction. Please see the revised introduction in page 1-3.
Point 5: Authors should clearly explain why older adults with dementia and chronic mental illness were grouped together as their target participants. Clinically, these two groups are very different. This should be properly address.
Response 5: In Japan, schizophrenia and dementia are treated at psychiatric hospitals. In addition, older patients with dementia and/or schizophrenia who have stable symptoms receive care at a facility for the elderly. This information was added in Page 6.
Point 6: Some easily observable editing mistakes in line 84-85.
Response 6: Thank you for your comments. We revised and added the heading for the clarity of the sentences in line 84-85.
Point 7: Authors should make clear what is the Framework of Compassionate Humanoid Health care Robots Communication and what paradigm it belongs to? what level of evidence had been yielded? And most importantly, any support of application of this framework to older adults with dementia and chronic mental illness.
Response 7: We revised the framework in page 3-4.
Point 8: Suggest authors should check for similarity (by either Veriguide orTurn-it-in), as I noted the writing style varies over the entire manuscript.
Response 8: Thank you very much. We checked the similarity index (result was 29%, and without references was 10%) using iThenticate. Please see attached results.
Point 9: Please write clearly how many subjects were recruited for this study. In line 32, authors marked 2, but in line 229, authors stated 6.
Response 9: There were four subjects in two cases on this article. One case involved two subjects. We revised the sentences in line 198.
Point 10: Please write clearly if Pepper will interact with one subject or one group of subjects in the stated procedure?
Response 10: Pepper interacted with two subjects in one case. This information was stated in the stated procedure in page 6.
Point 11: How can authors ensure the compliance and adherence of subjects when they interact with Pepper? This is a very questionable confounding factor.
Response 11: We added explanation sentences in page 6.
Point 12: Base on what inclusion and exclusion criteria did authors select their subjects? what types of schizophrenia? how chronic she was ?? based on what criteria that you chose ?? very unclear.
Response 12: Thank you very much. Inclusion and exclusion criteria are described in page 6 and 7. Also, the details of specific cases are described in page 7.
Point 13: What level of dementia did the subjects E and F were? unclear too.
Response 13: The details of specific cases are described page 8. We deleted the healthy subjects’ data, thus name of the subjects E and F were changed to C and D.
Point 14: Where are the results?? It seems this section is omitted in the entire manuscript.
Response 14: Thank you very much. The results were added after explanation of each case (Page 8 and 9).
Point 15: Authors should majorly revamp their discussion. The discussion in its current form just like another part of introduction. Suggest authors should follow the track of writing from Background (what knowledge had been identified and what knowledge gap should be addressed by authors), then Aim (I suppose they are working on Communication ?!). then Methods (with Peppers and AI ??), then Results (Even this is a case study, authors should report results in participants), then Discussion (discussion with result collected)
Response 15: The discussion was revised in page 10-11.
Point 16: Based on the present manuscript, I cannot seem how authors integrate AI and NLP at all. Can author reinforce this part in their methods?? Moreover, the conclusion is over-stretched with consideration of all the weakness as I mentioned from point 1-15.
Response 16: The conclusion was revised in page 12.
Point 17: To enrich the readability of the manuscript, English editing is a must for the revision.
Response 17: The manuscript has been revised.

Reviewer 2 Report
1- For the methodological part:
(a). define more precisely the patient characteristics of the cases compared to the reported case studies (even dementias can have various levels of severity and participation),
(b). define whether they had interacted with AI before or not, how much time did they spend this interaction.
2- Propose qualitative and quantitative assessment measures,
if not done in the study, for future studies
3- in the discussion, address and bring out the critical points with respect to the "replacement in care and relationship" of the Robot-AI instead of the person, from an ethical and social point of view.
Author Response
Point 1: For the methodological part: define more precisely the patient characteristics of the cases compared to the reported case studies (even dementias can have various levels of severity and participation).
Response 1: Thank you very much. Inclusion and exclusion criteria are described in page 6 and 7. Also, the details of specific cases are described page 7.
Point 2: For the methodological part: define whether they had interacted with AI before or not, how much time did they spend this interaction.
Response 2: Thank you for your comment. Subjects had interacted with Pepper several times because both settings have used Pepper before data collection. Time spent for interaction with Pepper in the study was approximately 20-30 minutes. This experiment was their first conversation experience with a Pepper. This information was added in page 6.
Point 3: Propose qualitative and quantitative assessment measures, if not done in the study, for future studies
Response 3: Thank you for giving us a good suggestion. We are thinking a new research design method is needed to respond to the complexities of clinical research problems involving sophisticated healthcare technologies particularly those involving the utility of healthcare robots in transactive engagements with human nurses. We are now conducting the novel research design human-robot interaction phenomena is challenging clinical nursing research methodology. It plans to submit the other journal. Thank you again.
Point 4: In the discussion, address and bring out the critical points with respect to the "replacement in care and relationship" of the Robot-AI instead of the person, from an ethical and social point of view.
Response 4: Thank you very much for your suggestion. In this study, we focus on exploring the issues of developing healthcare robots, therefore we did not address the replacement in care and relationship.

Round 2
Reviewer 1 Report
Developmental Issues of Healthcare Robots and Compassionate Care in Rehabilitation and Communication Activities of Older Adults with Dementia and Persons with Schizophrenia.
Thanks for effort of authors in revising the manuscript and this revision comes with a better quality and easier for reading. However, there are some writing issues that I have recommended authors to take actions but seems not effective enough.
- The title still far too busy for a lay-man to understand. What is the meaning of compassionate care in rehabilitation ? It is technically hard to read with a title with three “AND” in linking the three different concepts in a title.
- Developmental or Development Issue ?? As nouns the difference between development and developmental is that development is (uncountable) the process of developing; growth, directed change. I would choose “Development” (These types of English editing issues occurred throughout this entire revision and I solicited authors to seek English editing in my 1st response letter point 17. It seems my request haven’t been addressed.)
- Abstract is not well-revised too. “However, developmental issues exist. The purpose of this article is to explore the developmental issues of healthcare robots and compassionate care in rehabilitation and communication activities of persons with schizophrenia and older adults with dementia.” …. Developmental then developmental ??
- Introduction was majorly revised. However, it seems lacking of focus. Can authors show any literature support for their study on developing issues of healthcare robots ?? What is / are present compassionate care in rehabilitation ? what are the present communication activities of older adults with dementia and persons with Schizophrenia ?? As a researcher, authors need to formulate your research objectives, as you stated in abstract “The purpose of this article is to explore the developmental issues of healthcare robots and compassionate care in rehabilitation and communication activities of persons with schizophrenia and older adults with dementia.”
- Why social isolation suddenly dashed out after the “Significance of the study ” ?? Moreover, social isolation seems not being included as in your study objectives at all !!
- A lot of typos …. “Pychotherapy,” (line 77)… “the first conversational experience of the subjects” (line 202)
- Authors addressed my point 5 as “In Japan, schizophrenia and dementia are treated at psychiatric hospitals. In addition, older patients with dementia and/or schizophrenia who have stable symptoms receive care at a facility for the elderly. This information was added in Page 6.” … It seems authors still not aware the neuro-cognitive function and rehabilitation strategies in these two distinct groups are different. I suppose authors should read what is cognitive remediation and cognitive compensation. Nevertheless, what author supplemented in Page 6 was their inclusion criteria. They did not tell reader why these two groups were selected together.
- Authors tried to address my point 7. However, I still cannot see why “Framework of Compassionate Humanoid Healthcare Robots Communication” was used to govern this research on people with dementia and schizophrenia. Please provide more solid and evidence-based evidence of the captioned framework. I have to say, I spent more than half hours in searching but there is only a model “Development of Humanoid Healthcare Robots in Caring Science” on (https://connect.springerpub.com/content/sgrijhc/23/2/112)
- Authors addressed part of my point 9. “There were four subjects in two case studies. One case study involved two subjects”. How can authors control the potential bias then ? any methodological method ?? any analytical method ??
- Authors claimed addressed my point 11 in page 6. I cannot locate where they addressed despite I read Page 6 for 4 times. Compliance and adherence of subjects (particularly in older people with dementia) is crucial.
- Can authors provide more scientific and objective criteria for subjects’ screening. Say with the use of cognitive assessment like MoCA or psychiatric screening through NPI ? otherwise, it is hard for reader to believe an advance dementia older people can provide their consent and can merely get rid of cognitive problem
- I noted authors added the “Results”, but they did not tell readers what is/are the measurement of the effectiveness beforehand. Moreover, who performed the rating? How to ensure the objectivity ?? All these important parts are missing!!
- Discussion should be based upon Results yielded. Conclusion should summarize the findings of whole study, either support the literature or rebut with evidence generated from the study. However, most of content in the conclusions like “Findings from two cases identified issues regarding robot`s gaze and listening function. For robot to express compassionate care to older adults, future development of healthcare robots need improvement in its ability to communicate with more than one person or to group of people with appropriate eye contact/gaze to who is talking. In addition, with expected function as communication partner to older people, healthcare robots need to be designed to have capability of listening low voice Two case examples of social conversation between robots and older adults, including those diagnosed with schizophrenia and dementia” is repeated from Discussion. Most importantly, I think both parts are over-stretched.
- It is obvious no professional English editing has been performed.